# Quercetin’s Potential in MASLD: Investigating the Role of Autophagy and Key Molecular Pathways in Liver Steatosis and Inflammation

**DOI:** 10.3390/nu16223789

**Published:** 2024-11-05

**Authors:** Ioannis Katsaros, Maria Sotiropoulou, Michail Vailas, Emmanouil Ioannis Kapetanakis, Georgia Valsami, Alexandra Tsaroucha, Dimitrios Schizas

**Affiliations:** 1First Department of Surgery, National and Kapodistrian University of Athens, Laikon General Hospital, 17 AgiouThoma Str., Athens 11527, Greece; marosotiropoulou@gmail.com (M.S.); mike_vailas@yahoo.com (M.V.); schizasad@gmail.com (D.S.); 2Department of Thoracic Surgery, National and Kapodistrian University of Athens, Attikon University Hospital, Athens12462, Greece; emmanouil.kapetanakis@gmail.com; 3Laboratory of Biopharmaceutics-Pharmacokinetics, Department of Pharmacy, School of Health Sciences, National and Kapodistrian University of Athens, Athens 15774, Greece; valsami@pharm.uoa.gr; 4Laboratory of Experimental Surgery, Faculty of Medicine, Democritus University of Thrace, Alexandroupolis 68100, Greece; atsarouc@med.duth.gr

**Keywords:** quercetin, NAFLD, MASLD, autophagy, Beclin1, LC3A, SQSTM1, CD36, Perilipin3

## Abstract

Metabolic dysfunction-associated fatty liver disease (MASLD) is a widespread liver disorder characterized by excessive fat accumulation in the liver, commonly associated with metabolic syndrome components such as obesity, diabetes, and dyslipidemia. With a global prevalence of up to 30%, MASLD is projected to affect over 100 million people in the U.S. and 20 million in Europe by 2030. The disease ranges from Steatotic Lived Disease (SLD) to more severe forms like metabolic dysfunction-associated steatohepatitis (MASH), which can progress to cirrhosis and hepatocellular carcinoma. Autophagy, a cellular process crucial for lipid metabolism and homeostasis, is often impaired in MASLD, leading to increased hepatic lipid accumulation and inflammation. Key autophagy-related proteins, such as Beclin1, LC3A, SQSTM1 (p62), CD36, and Perilipin 3, play significant roles in regulating this process. Disruption in these proteins contributes to the pathogenesis of MASLD. Quercetin, a natural polyphenolic flavonoid with antioxidant and anti-inflammatory properties, has promising results in mitigating MASLD. It may reduce hepatic lipid accumulation, improve mitochondrial function, and enhance autophagy. However, further research is needed to elucidate its mechanisms and validate its therapeutic potential in clinical settings. This underscores the need for continued investigation into autophagy and novel treatments for MASLD.

## 1. Introduction

Non-alcoholic fatty liver disease (NAFLD) is a major public health disease defined by liver fat accumulation with concomitant cardiometabolic risk factors and the absence of harmful alcohol intake [1]. The European Association for the Study of the Liver (EASL) and the American Association for the Study of Liver Diseases (AASLD) adopted the term metabolic dysfunction-associated fatty liver disease (MASLD) in order to better reflect its pathophysiologic background and alleviate the stigma often associated with it [2,3]. MASLD has a global prevalence of up to 30% and is projected to affect over 100 million people in the United States and 20 million in Europe by 2030 [4,5]. It is a part of the metabolic syndrome and is associated with diabetes mellitus type 2, dyslipidemia, and obesity [6,7,8]. The severity of MASLD spans from Steatotic Lived Disease (SLD) to metabolic dysfunction-associated steatohepatitis (MASH), with potential progression to liver cirrhosis and hepatocellular carcinoma (HCC) [1,2,9,10]. Early diagnosis, along with therapeutic interventions and lifestyle modifications, can potentially reverse MASLD in its initial stages, whereas the later stages are marked by irreversible liver damage, often necessitating liver transplantations [11].

The development of MASLD was traditionally understood through the “two-hits” pathophysiological model. In this framework, the first “hit” involves the accumulation of lipids within hepatocytes, leading to liver steatosis. This is largely driven by an imbalance between the input and output of fatty acids within the liver cells, often associated with metabolic disturbances such as central obesity and insulin resistance [12,13]. The hepatic lipid overload creates a vulnerable state, priming the liver for further injury. The “second hit” is thought to occur when additional factors, including inflammatory cytokines, oxidative stress, bacterial endotoxins, and mitochondrial dysfunction, exacerbate liver damage, causing inflammation, hepatocellular injury, and eventually fibrosis [14,15].

However, more recent insights have shifted the paradigm from the “two-hits” model toward a more nuanced understanding known as the “multiple parallel hits” hypothesis. This hypothesis suggests that rather than a sequential process, various factors act simultaneously to contribute to the progression of MASLD. These factors include genetic predispositions, alterations in gut microbiota, increased circulating free fatty acids, and adipokine imbalances, which together lead to insulin resistance, oxidative stress, and chronic inflammation. As a result, MASLD is now recognized as a multifactorial disease, where metabolic, environmental, and genetic elements collectively trigger the development and progression of liver steatosis, inflammation, and fibrosis [16]. This hypothesis better explains the complexity of the disease and the heterogeneity observed in its clinical manifestations, emphasizing that liver damage can occur at varying levels and stages depending on the interplay of these contributing factors.

The development of both experimental and clinical studies is crucial to fully understand the underlying pathophysiological pathways of MASLD. Novel therapeutic approaches and agents targeting these pathways could potentially reverse liver steatosis, inflammation, and hepatocellular injury.

## 2. The Role of Quercetin in MASLD

Several pharmacological treatments have been proposed against MASLD, but they have shown limited efficacy and long-term results [17]. Natural polyphenols are a heterogeneous class of polyphenolic compounds, are used for the treatment of metabolic disorders, and have been shown to have a hepatoprotective action [18]. This action is mainly a result of both the increased lipid breakdown in the liver and the reduction in fibrosis lipolysis [19]. Thus, oxidative stress and the subsequent hepatocellular damage are reduced [19].

Quercetin (QUE) is a naturally occurring polyphenolic flavonoid widely recognized for its broad range of beneficial biological properties, including anti-inflammatory, antioxidant, anti-apoptotic, and immunoprotective effects [20]. Numerous studies have demonstrated quercetin’s hepatoprotective capabilities, particularly in the context of liver diseases, making it a promising candidate for therapeutic use. However, the exact pathophysiological mechanisms through which quercetin exerts its protective effects on MASLD remain not fully elucidated [20,21,22,23]. Some research suggests that quercetin may help reduce hepatic lipid accumulation, improve mitochondrial function, and modulate oxidative stress, all of which play a role in MASLD pathogenesis. Autophagy plays a critical role in maintaining lipid homeostasis in hepatocytes, primarily through the process of lipophagy—the selective degradation of lipid droplets. Impaired autophagy in MASLD contributes to excessive lipid accumulation, which exacerbates inflammation and liver damage. Studies show that downregulated autophagy markers, including Beclin1 and LC3A, correlate with increased hepatic lipid droplet accumulation [24,25]. Additionally, quercetin may attenuate liver inflammation by regulating cytokine production and mitigating the harmful effects of oxidative stress-induced apoptosis within hepatocytes [20,21,22,23].

Alongside QUE, several other polyphenols are under investigation for their therapeutic potential in MASLD due to their anti-inflammatory, antioxidant, and lipid-modulating properties [26,27]. Resveratrol, a polyphenol found in grapes, has shown promise in reducing hepatocellular damage, as it activates the AMPK pathway and sirtuin-1 (SIRT1), leading to reduced inflammation and reduced oxidative stress [28,29]. Curcumin, a compound derived from turmeric, exhibits anti-inflammatory and antifibrotic properties. By modulating pathways such as NF-κB and TGF-β, curcumin reduces hepatic inflammation and fibrosis, making it another potential therapeutic agent in MASLD [26,30]. Epigallocatechin gallate (EGCG), a green tea polyphenol, has also demonstrated hepatoprotective effects. EGCG decreases lipogenesis and oxidative stress through the activation of AMPK and inhibition of lipogenic enzymes like SREBP1 [31]. These polyphenols, similar to QUE, show multi-targeted effects on lipid metabolism, inflammation, and oxidative stress, highlighting the potential of polyphenol-based interventions for managing MASLD. Future studies comparing the efficacy and mechanisms of these compounds could further clarify their roles as natural treatments in MASLD.

Given these properties, quercetin shows potential as a novel therapeutic agent for managing MASLD, offering a natural and multifaceted approach to addressing both liver steatosis and its associated complications. Further research is needed to confirm its efficacy and underlying mechanisms in human models.

## 3. Current Evidence

Several experimental animal studies have shown the hepatoprotective action of natural polyphenols, including QUE against liver steatosis and MASLD, but their mechanism of action is not yet elucidated. Yang et al. suggested the protective action of QUE in a mice MASLD model by restoring lipid homeostasis through the downregulation of the mTOR/YY1 signaling pathway [32]. He et al. investigated the effect of the famous traditional Chinese shengi pill, whose active compound is QUE, on a MASH rat model and concluded that liver injury and lipid metabolism were significantly improved after its administration by reducing the effect of pro-inflammatory cytokines (IL-6, TNF-a) [33]. Ying et al. reached the same conclusion showing that oral intake of QUE for 14 days in a high-fat-diet-induced MASH rat model significantly reduced liver lipid accumulation by lowering pro-inflammatory cytokines TNF-a and IL-6 [34]. Furthermore, Ma et al. revealed that QUE attenuated steatohepatitis by inhibiting the activated NLRP2 inflammasome through suppression of the expression of HSP90 [35]. An amelioration of MASH was also reported by Macrolin et al., who also showed an attenuation of several pro-fibrotic and pro-inflammatory pathways in mice fed with QUE [36]. Finally, Panchal et al. also reported an improvement in MASLD induced in high-fat fed rats after 8 weeks of QUE by downregulating the NF-kB pathway, a factor stimulating inflammation, and upregulating Nrf2, a major protective factor against oxidative stress [37].

MASLD progression is influenced by multiple pathways, each of which quercetin may modulate to reduce lipid accumulation, inflammation, and fibrosis (Figure 1). The AMPK pathway plays a central role in energy regulation and induces mitophagy, helping maintain mitochondrial health and reducing oxidative stress. Studies indicate that quercetin activates AMPK, thereby supporting mitophagy and mitigating mitochondrial dysfunction in MASLD [22]. The SREBP1/FAS pathway, which regulates lipid synthesis, is often upregulated in MASLD, contributing to excess lipid accumulation. Quercetin inhibits SREBP1 and FAS expression, decreasing lipogenesis and lipid accumulation [38]. Another key pathway, PI3K/AKT, is involved in cell growth and survival; quercetin modulates this pathway, attenuating hepatic inflammation and fibrosis [32]. Additionally, quercetin affects FXR/TGR signaling, enhancing bile acid metabolism and reducing lipid overload [18]. Finally, the TGF-β1/Smads pathway, a major driver of fibrosis in MASLD, is downregulated by quercetin, which decreases fibrotic markers and may slow disease progression [35]. Together, these pathways highlight the multifaceted role of quercetin in addressing MASLD’s complex pathophysiology.

QUE has been shown to modulate autophagy through the mTOR pathway, influencing lipid metabolism and reducing hepatic inflammation. By activating autophagic processes, quercetin potentially restores lipid homeostasis, alleviating steatosis and inflammation in MASLD [32]. In addition to lipophagy, mitophagy—a selective form of autophagy that removes dysfunctional mitochondria—is relevant to MASLD pathogenesis. Mitochondrial dysfunction contributes to oxidative stress and inflammation in MASLD. Furthermore, QUE activates AMPK-mediated mitophagy and regulates Frataxin-mediated PINK1–Parkin-dependent mitophagy, which can help reduce mitochondrial damage and oxidative stress, thereby potentially mitigating the progression of MASLD [21,39]. Finally, through the TGF-β1/Smads and PI3K/Akt pathway, QUE exerts antifibrotic effects that are critical for slowing or preventing fibrosis progression in MASLD. By modulating this pathway, QUE could serve as an adjunct therapy to reduce liver fibrosis associated with chronic MASLD [36,40].

Clinical studies investigating the effect of QUE on MASLD are limited, but they have reached similar conclusions to experimental studies. Prysyazhnyuk and Voloshyn recruited 41 MASLD patients and revealed a significant decrease in the levels of total cholesterol, triacylglycerol, and liver function tests following two weeks of QUE, showing a potential protective effect of QUE [41]. Pasdar et al. enrolled 90 patients suffering from MASLD and randomly assigned them to either a QUE or placebo group for 12 weeks. Patients receiving QUE had a significant increase in red blood cell count, thus reversing the effect of oxidative stress [42].

## 4. Autophagy and MASLD

Autophagy is a crucial cellular process responsible for maintaining homeostasis by degrading and recycling damaged organelles and proteins. It plays a vital role in hepatocellular lipid metabolism, which is essential for liver health and has been implicated in the pathogenesis of MASLD [43]. In the setting of MASLD, autophagy is frequently downregulated, contributing to increased lipid accumulation in hepatocytes. Evidence indicates that impaired autophagy in the fatty liver occurs primarily due to the disruption of autophagosome–lysosome fusion, leading to the accumulation of autophagosomes and ineffective degradation of intracellular lipids [44]. This defective autophagic flux contributes to impaired lipophagy, further promoting hepatic steatosis and the development of MASLD [45].

A critical component of this process involves key autophagy proteins such as Beclin1 and LC3A. Beclin1 plays a major role in initiating autophagy by forming complexes that promote autophagosome formation. LC3A is responsible for the elongation and closure of autophagosomes. Research by Zhao et al. has demonstrated that hepatic steatosis results in the downregulation of both Beclin1 and LC3A, leading to decreased autophagic activity and the accumulation of lipid droplets in hepatocytes [25]. Another critical autophagy-related protein is SQSTM1 (p62), which serves as a marker of impaired autophagy. SQSTM1 binds to ubiquitinated proteins, facilitating their degradation through autophagy. When autophagy is disrupted, SQSTM1 accumulates, signaling defective autophagic processes and contributing to the further progression of MASLD [45,46].

In addition to these key autophagic proteins, other factors such as CD36 and Perilipin 3 also play significant roles in the development of MASLD [47,48]. CD36 is a fatty acid transporter that facilitates the uptake of lipids into hepatocytes. In MASLD, CD36 is often upregulated, contributing to increased lipid accumulation and subsequent liver damage [47]. Perilipin 3, a regulator of lipid droplet dynamics, is essential for the mobilization of lipids within hepatocytes. Dysregulation of Perilipin 3 can impair lipid metabolism, further aggravating hepatic steatosis and inflammation in MASLD [48].

Autophagy serves as a protective mechanism in liver cells, regulating lipid homeostasis and preventing excess lipid buildup. However, in MASLD, its downregulation leads to a cascade of metabolic disturbances that promote the progression from simple steatosis to more severe liver conditions, including metabolic dysfunction-associated steatohepatitis (MASH), fibrosis, and cirrhosis [24,46,49]. Further research is needed to fully elucidate the molecular mechanisms behind autophagy impairment in MASLD, particularly the roles of Beclin1, LC3A, SQSTM1, CD36, and Perilipin 3. A deeper understanding of these processes may lead to novel therapeutic strategies targeting autophagy to reverse hepatic steatosis and prevent disease progression.

## 5. Conclusions and Implications for Future Research

MASLD is a prevalent and growing liver disease affecting a substantial proportion of the global population. While some progress has been made in understanding its pathophysiology, the mechanisms driving liver steatosis, oxidative stress, impaired autophagy, and mitochondrial dysfunction in hepatocytes remain unclear. Unraveling these pathways is essential for developing therapeutic strategies that can halt disease progression or potentially reverse liver damage.

Flavonoids like quercetin have demonstrated hepatoprotective properties, showing promise as potential treatments for MASLD. Specifically, quercetin’s antioxidant and anti-inflammatory effects have been supported by several experimental studies, suggesting it may help reduce liver injury and steatosis in MASLD. Autophagy, a critical cellular process that removes damaged organelles and excess lipids, plays a significant role in maintaining liver homeostasis. Impaired autophagy has been implicated in the progression of MASLD, particularly in the accumulation of lipid droplets and the promotion of liver inflammation. Quercetin’s potential to modulate autophagy and promote lipophagy could be a key mechanism in reducing hepatic steatosis and inflammation. However, further investigation is needed to fully understand the pathophysiological mechanisms underlying its effects.

Current research on quercetin’s role in autophagy, lipophagy, and the alleviation of hepatic steatosis and inflammation is limited, creating uncertainty about its efficacy against MASLD. Additional experimental and clinical studies are required to clarify the role of quercetin in MASLD.

## Figures and Tables

**Figure 1 nutrients-16-03789-f001:**
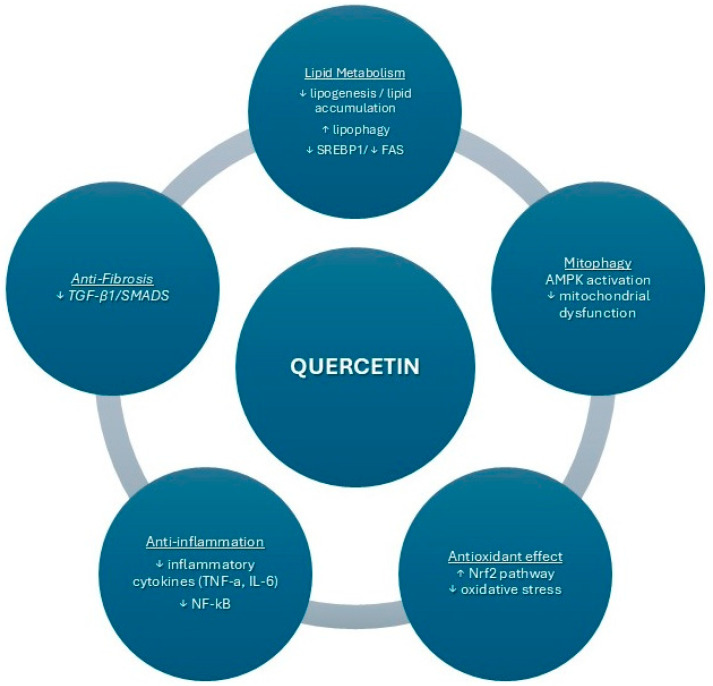
The effect of Quercetin on MASLD (metabolic dysfunction-associated fatty liver disease).

## Data Availability

Data sharing is not applicable.

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
