# Peer review of "Quercetin’s Potential in MASLD: Investigating the Role of Autophagy and Key Molecular Pathways in Liver Steatosis and Inflammation"

_nutrients, 2024, doi:10.3390/nu16223789_

Round 1

Reviewer 1 Report

Comments and Suggestions for Authors

The review manuscript focused on summarizing the role of autophagy and molecular pathways and possible impact of a natural polyphenol, quercetin in NAFLD mediated fatty liver and inflammation. However, this reviewer has raised following major concerns regarding the manuscript:

1)      There is at least one published article that describes the role of quercetin and NAFLD (doi: 10.4103/sjg.sjg_249_21). In this manuscript, authors intended to relate autophagy and key pathways involved in NAFLD and possible impact of quercetin in this liver disease. However, the description about autophagy and its impact on NAFLD were insufficient.

2)      Authors ignored the relationship between NAFLD mediated autophagy and quercetin.

3)      They tried to discuss about the lipophagy process but never used the term until the conclusion.

4)      Authors should discuss about hepatic mitophagy and its putative relationship with NAFLD and quercetin.

5)      The title of the manuscript indicated to discuss the key pathways in NAFLD mediated liver steatosis in connection with quercetin. However, the description of these possible major pathways was largely lacking. The should include a) AMPK-mediated hepatic mitophagy, b) SREBP1/FAS, c) Cdkn1a, d) PI3K/AKT pathway, e) FXR/TGR signaling pathway and f) TGF-β1/Smads.

6)      In addition, authors need to include a discussion on the effect of quercetin on hepatic fibrosis.

7)      There should be a diagram depicting the pathways and indication of putative action of quercetin in it.

Author Response

Comment 1: There is at least one published article that describes the role of quercetin and NAFLD (doi: 10.4103/sjg.sjg_249_21). In this manuscript, authors intended to relate autophagy and keypathways involved in NAFLD and possible impact of quercetin in this liver disease. However, the description about autophagy and its impact on NAFLD were insufficient.

Response 1: Thank you for this valuable comment. We acknowledge that additional information on the role of autophagy in NAFLD is critical. In response, we have expanded our discussion on autophagy, emphasizing its role in regulating lipid metabolism through processes like lipophagy and mitophagy and their contribution to hepatic inflammation.

Comment 2: Authors ignored the relationship between NAFLD mediated autophagy and quercetin.

Response 2: We appreciate this point. We have now included additional discussion on the potential mechanisms by which quercetin may modulate autophagy at the relevant section.

Comment 3:  They tried to discuss about the lipophagy process but never used the term until the conclusion.

Response 3: Thank you for highlighting this. We have revised the manuscript to introduce the term “lipophagy” earlier in the discussion on autophagy. This clarification should help readers understand the relevance of lipophagy in hepatic lipid metabolism and its connection to NAFLD pathophysiology.

Comment 4: Authors should discuss about hepatic mitophagy and its putative relationship with NAFLD and quercetin.

Response 4: We have expanded the discussion to include hepatic mitophagy and its role in NAFLD. We address how mitochondrial dysfunction in NAFLD can be ameliorated by quercetin through mitophagy induction.

Comment 5: The title of the manuscript indicated to discuss the key pathways in NAFLD mediated liver  steatosis in connection with quercetin. However, the description of these possible major         pathways was largely lacking. The should include a) AMPK-mediated hepatic mitophagy,               b) SREBP1/FAS, c) Cdkn1a, d) PI3K/AKT pathway, e) FXR/TGR signaling pathway and                        f) TGF-β1/Smads.

Response 5: Thank you for your input. We expanded our manuscript to include detailed descriptions of these key molecular pathways.

Comment 6: In addition, authors need to include a discussion on the effect of quercetin on hepatic fibrosis.

Response 6: Thank you for your comment. We incorporated a section discussing quercetin’s potential anti-fibrotic effects.

Comment 7: There should be a diagram depicting the pathways and indication of putative action of quercetin in it.

Response 7: We prepared a diagram (Figure 1) illustrating the major pathways involved in NAFLD.

Reviewer 2 Report

Comments and Suggestions for Authors

Autophagy and Key Molecular Pathways in Liver Steatosis and Inflammation

Nonalcoholic fatty liver disease (NAFLD) is the most common form of chronic liver disease, affecting almost one-third of the general population and 75% of obese patients with type 2 diabetes. The aim of this article is to review the current evidence concerning the role of quercetin, a natural compound and flavonoid, and its possible therapeutic effects on this modern-day disease. The article is interesting as a topic, but it is extremely concise. There are similar articles from 2015, 2021, 2022 (Sotiropoulou M, Katsaros I, Vailas M, Lidoriki I, Papatheodoridis GV, Kostomitsopoulos NG, Valsami G, Tsaroucha A, Schizas D. Nonalcoholic fatty liver disease: The role of quercetin and its therapeutic implications. Saudi J Gastroenterol. 2021 Nov-Dec;27(6):319-330; Gnoni A, Di Chiara Stanca B, Giannotti L, Gnoni GV, Siculella L, Damiano F. Quercetin Reduces Lipid Accumulation in a Cell Model of NAFLD by Inhibiting De Novo Fatty Acid Synthesis through the Acetyl-CoA Carboxylase 1/AMPK/PP2A Axis. Int J Mol Sci. 2022; 

Quercetin Alleviates High-Fat Diet-Induced Oxidized Low-Density Lipoprotein Accumulation in the Liver: Implication for Autophagy Regulation, 2015). It needs clear improvement with new mechanisms and images,text restructuring.

Author Response

Comment 1: Nonalcoholic fatty liver disease (NAFLD) is the most common form of chronic liver disease, affecting almost one-third of the general population and 75% of obese patients with type 2 diabetes. The aim of this article is to review the current evidence concerning the role of quercetin, a natural compound and flavonoid, and its possible therapeutic effects on this modern-day disease.

The article is interesting as a topic, but it is extremely concise. There are similar articles from 2015, 2021, 2022 (Sotiropoulou M, Katsaros I, Vailas M, Lidoriki I, Papatheodoridis GV, Kostomitsopoulos NG, Valsami G, Tsaroucha A, Schizas D. Nonalcoholic fatty liver disease: The role of quercetin and its therapeutic implications. Saudi J Gastroenterol. 2021 Nov-Dec;27(6):319-330; Gnoni A, Di Chiara Stanca B, Giannotti L, Gnoni GV, Siculella L, Damiano F. Quercetin Reduces Lipid Accumulation in a Cell Model of NAFLD by Inhibiting De Novo Fatty Acid Synthesis through the Acetyl-CoA Carboxylase 1/AMPK/PP2A Axis. Int J Mol Sci. 2022; Quercetin Alleviates High-Fat Diet-Induced Oxidized Low-Density Lipoprotein Accumulation in the Liver: Implication for Autophagy Regulation, 2015).

It needs clear improvement with new mechanisms and images , text and restructuring.

Response 1: Thank you for your comments and valuable input. We elaborated on recent mechanisms identified in NAFLD, such as the role of quercetin in mitophagy and lipophagy and key molecular pathways. Additionally, we included an illustrative diagram to provide a visual summary of the discussed pathways.

Reviewer 3 Report

Comments and Suggestions for Authors

The authors aimed to consider the role of quercetin in the course of NAFLD. I suggest below-listed improvements.

  1. The term of NAFLD is not up to date. Instead of it the name of MASLD should be included.

  2. The topic is too narrow and the numer of references is definitely too low. From my perspective the context of other possible agents in the treatment of this pathology should be added.

Author Response

Comment 1: The term of NAFLD is not up to date. Instead of it the name of MASLD should be included.

Response 1: Thank you for your comment. We acknowledge the updated terminology and have revised the manuscript to refer to MASLD as the preferred term, in line with the latest guidelines

Comment 2: The topic is too narrow and the number of references is definitely too low. From my perspective the context of other possible agents in the treatment of this pathology should be added.

Response 2: Thank you for your input. The aim of our manuscript was to elucidate the role of quercetin on NAFLD and its implication in related key autophagy pathways.

Round 2

Reviewer 1 Report

Comments and Suggestions for Authors

This reviewer is satisfied with the current content of the manuscript